# Aflatoxicosis Dysregulates the Physiological Responses to Crowding Densities in the Marine Teleost Gilthead Seabream (*Sparus aurata*)

**DOI:** 10.3390/ani11030753

**Published:** 2021-03-09

**Authors:** Andre Barany, Juan Fuentes, Gonzalo Martínez-Rodríguez, Juan Miguel Mancera

**Affiliations:** 1Department of Biology, Faculty of Marine and Environmental Sciences, Instituto Universitario de Investigación Marina (INMAR), Campus de Excelencia Internacional del Mar (CEI MAR), University of Cádiz, Puerto Real, 11519 Cádiz, Spain; juanmiguel.mancera@uca.es; 2Centre of Marine Sciences (CCMar), Universidade do Algarve, Gambelas, 8005-139 Faro, Portugal; jfuentes@ualg.pt; 3Instituto de Ciencias Marinas de Andalucía, Consejo Superior de Investigaciones Científicas (ICMAN-CSIC), Puerto Real, 11519 Cádiz, Spain; gonzalo.martinez@csic.es

**Keywords:** *sparidae*, stress, aquaculture, stocking density, endocrine-related genes, energetic metabolism, mycotoxin

## Abstract

**Simple Summary:**

A fungal toxin, aflatoxin B1 (AFB1), undermines growth and stress axes of gilthead seabream (*Sparus aurata*) with depletion of somatic carbohydrate and lipid reservoirs. The present study assessed the physiological consequences of high stocking density versus low stocking density in seabream juveniles, which had previously been fed with AFB1 supplementation. These stressors are likely to converge by inferring animal welfare and economic profitability in the food animal industry. Interestingly, AFB1 seems to cause physiological and molecular dysfunction in response to overcrowding densities. Our results might be relevant to elucidate a potential risk for fish farming that is often overlooked.

**Abstract:**

Several studies in fish have shown that aflatoxin B1 (AFB1) causes a disparity of species-dependent physiological disorders without compromising survival. We studied the effect of dietary administration of AFB1 (2 mg AFB1 kg^−1^ diet) in gilthead seabream (*Sparus aurata*) juveniles in combination with a challenge by stocking density (4 vs. 40 g L^−1^). The experimental period duration was ten days, and the diet with AFB1 was administered to the fish for 85 days prior to the stocking density challenge. Our results indicated an alteration in the carbohydrate and lipid metabolites mobilization in the AFB1 fed group, which was intensified at high stocking density (HSD). The CT group at HSD increased plasma cortisol levels, as expected, whereas the AFB1-HSD group did not. The *star* mRNA expression, an enzyme involved in cortisol synthesis in the head kidney, presented a ninefold increase in the AFB1 group at low stocking density (LSD) compared to the CT-LSD group. Adenohypophyseal *gh* mRNA expression increased in the AFB1-HSD but not in the CT-HSD group. Overall, these results confirmed that chronic AFB1 dietary exposure alters the adequate endocrinological physiological cascade response in *S. aurata*, compromising the expected stress response to an additional stressor, such as overcrowding.

## 1. Introduction

The primary stressors in aquaculture practices are transportation, handling, and crowding, causing higher bioenergy expenditure by disturbing animal welfare [1]. Furthermore, space optimization by increasing stocking density improves economic profitability and, as with industrial terrestrial livestock farms, fish are often kept in unnaturally crowded conditions. Overcrowding is a stressful condition that activates the stress control system, inducing a primary stress response with enhanced catecholamines and cortisol release into the bloodstream. If this situation persists, a secondary stress response will occur by altering metabolism, hydromineral balance, and/or the immune system [2,3]. Ultimately, the effects will result in a tertiary stress response, potentially evoking impaired growth, reproductive outcomes, ability to overcome new stress, immune capacity alterations, abnormal behavior, and/or death [4,5].

Stress is a disruption of homeostasis resulting from the effects of biotic and/or abiotic factors, and the consequences depend on the type, duration, and intensity of the stressor [1,6]. The hypothalamic–pituitary–inter-renal (HPI) axis, which is stimulated in stress situations, releases hypothalamic corticotropin-releasing hormone (CRH), as well as the CRH-binding protein (CRHBP) and thyrotropin-releasing hormone (TRH). Subsequently, CRH stimulates adrenocorticotropic hormone (ACTH) release, which is derived from proopiomelanocortins (POMCs). These POMCs are produced in the adenohypophyseal corticotropic cells. The adenohypophyseal gland (or pituitary) acts as the interface between the brain and the endocrine systems to produce a coordinated physiological response [7,8]. In particular, two POMCs have been described in the *S. aurata* pituitary as part of the endocrine stress response—(i) *pomca1*, responsible for alpha-melanocyte-stimulating hormone (α-MSH) synthesis, and (ii) *pomca2*, responsible for ACTH production [9,10]. α-MSH has been implicated in some fish species in chronic stress responses [11,12]. Additionally, growth hormone (GH), also produced in the pituitary, is a well-known regulator of growth and metabolism [13,14]. However, its involvement in fish’s stress process has not yet been elucidated [15,16,17]. Ultimately, ACTH activates head kidney inter-renal cells to produce and release cortisol approximately five minutes later [7,10,18].

Cortisol is the primary corticosteroid in teleosts and the final messenger of the HPI axis [2,19]. Its circulating levels are implicated in a wide range of body functions, such as osmoregulation, metabolism, immune response, reproduction, and behavior [5]. In this sense, the corticosteroid component of the chronic stress-response focuses on long-term acclimation. Therefore, it provides a longer-lasting energy supply by modulating gluconeogenesis and lipolysis. This modification is achieved mainly by energy substrate redistribution and water retention [20,21,22]. Cortisol has also been demonstrated to increase liver protein synthesis, restricting its formation in muscles [23].

It has previously been shown that gilthead seabream exposure to an ethological stressful situation, such as crowding densities, induces the classical primary and secondary stress responses [3,8,24,25]. Overall, this species’ physiological responses to density stress include an increase in plasma cortisol, glucose, lactate, and triglycerides in order to cope with a higher bioenergy expenditure [19,25]. Moreover, gene transcription and corticosteroids released into the blood related to the HPI axis are up-regulated [10,26]. Remarkably, a prolonged inter-renal activity might cause a down-regulation of the involved components by negative feedback, which ultimately might desensitize to additional stressors. As a result, the fish’s physiological functions likely protect themselves from exhaustion, which can result in mortality [19].

On a large scale, aquaculture production is made profitable by rapid fish growth, which is heavily influenced by fish feeding and overall animal welfare. In contrast, its primary expense is the commercial fish feeds. The global decline of wild fish stocks for fishmeal production increases commercial fish feed costs [27,28], which, in turn, leads to a need to optimize and increase productivity by incorporating mixtures of plant-based ingredients and oils in feed formulations [12]. These plant-based fish feeds are successful in southern European fish farming, such as for the European sea bass and the gilthead seabream [29,30]. Although the general inclusion of more sustainable and cheaper ingredients is guaranteed to improve the production in the aquaculture sector, new obstacles associated with agricultural commodities, pre- and post-harvest, have arisen [31,32,33].

In this regard, mycotoxins found in fish feeds are an increasing problem [34,35,36,37]. Mycotoxins are low-molecular-weight compounds produced by some fungus strains as secondary metabolites [38]. They are a major issue if the stock storing conditions are inadequate, i.e., temperature, humidity, sealing [39]. However, some staple crops might present detectable mycotoxin levels, even when storing conditions are adequate [32]. Moreover, it is expected that their occurrence will increase linked to climate change conditions [40], especially those favorable to the growth of *Aspergillus* spp. and *Penicillium* spp.

Among the mycotoxins, aflatoxin B1 (AFB1) is the most hazardous, classified as a Group I carcinogen [41]. Its pathological clinical manifestations in vertebrates include immunosuppression, hepatotoxicity, genotoxicity, carcinogenic and teratogenic effects [42,43,44]. In mammals, aflatoxicosis induces, inter alia, carcinogenesis, and subsequent tumoral activity. Note that tumoral activity is correlated with the overproduction of reactive oxygen species [45,46]. In particular, it has been found that chronic ingestion is likely to be life-threatening at concentrations in the range of 0.02 to 8 mg/kg, 0.5 to 1.5 mg/kg, and 0.3 to 0.6 mg/kg of body mass, for humans, dogs, and cats, respectively [47,48].

Conversely, the apparent lack of clinical signs of aflatoxicosis in fish compared to terrestrial livestock [32] often results in this problem being overlooked for farmed fish. The negative impact of AFB1 in farmed fish and shrimps impairs growth, induces tissue damage [49,50] but may also cause dysregulation of intestinal microbiota [51]. Moreover, some behavioral abnormalities have also been observed, which are commonly associated with tertiary stress responses [5,50]. The toxicity of AFB1 is not only dose-dependent but also depends on the duration of exposure, as well as animal species, sex, age, and metabolism [52,53,54]. The worldwide regulation for overall aflatoxin levels for feed and feed ingredients lies in the range of 4–30 µg/kg [34]. Notably, the European Union has the strictest restriction for direct human consumption—2–4 µg/kg [55]. However, no specific regulations and guidelines are available for the aquaculture sector [32].

Overall, understanding the physiological responses to stressors recurrently encountered in aquaculture production might be relevant in stress management to maintain healthy fish stocks and, subsequently, optimize the production. In this regard, we focused on the culture density stressor and the occurrence of AFB1 in fish feeds. The gilthead seabream (*Sparus aurata*) is an important aquaculture species that has been intensively farmed for decades in the Mediterranean region [56,57]. In a previous study of this species, dietary AFB1 impaired growth, affecting carbohydrate and lipid energetic metabolites, accompanied by histopathological alterations in several tissues (liver, kidney, and spleen). Moreover, the expression of several endocrine players of HPI and growth axes were altered [50].

Therefore, in this study, we aimed to experimentally simulate a convergence of two different stressors in the gilthead seabream (*Sparus aurata*) throughout a previous chronic dietary AFB1 [50] followed by a challenge by stocking density (4 vs. 40 g L^−1^) [8,26,58,59,60]. Specifically, we compared the molecular and physiological responses to overcrowding stress in order to characterize specific dysregulations resulting from dietary aflatoxicosis.

## 2. Materials and Methods

### 2.1. Experimental Fish Feed

A commercial fish feed (Skretting, Burgos, Spain) was used as a basis for the experimental diets to provide all essential nutritional requirements for the gilthead seabream (57% crude protein, 18% crude fat, 10% ash, 1.6% phosphorus, and 19.5 MJ kg^−1^ digestible energy). The inclusion of AFB1 (Sigma A6636, St. Louis, MO, USA) at levels of 0 (control, CT) and 2 AFB1 mg kg^−1^ fish feed was performed after grinding and pelletization. Prof. Francisco Javier Moyano’s Research group, from the University of Almería, Spain, developed the experimental aquafeeds by extrusion at the Experimental Feed Service. Fish were previously fed at 1.5% of total biomass for 85 consecutive days in three equal portions throughout daylight, as reported by Barany et al. [50], before the experimental overcrowding challenge.

### 2.2. Animals and Experimental Designs

*Sparus aurata* juveniles were purchased from the local fish farm CUPIBAR SL (Cádiz, Spain). Fish (*n* = 80; 176.79 ± 2.52 g body mass, and 19.16 ± 0.09 cm fork length) were homogeneously distributed as groups of 10 individuals in eight 500 L-fiber glass tanks. They were kept with continuous aeration in a flow-through system, natural photoperiod (June 2017) for our latitude (36°31′44″ N) and constant temperature (19–20 °C) at the Servicio Central de Investigación en Cultivos Marinos (SCI-CM, CASEM, University of Cádiz, Spain; Operational Code REGA ES11028000312). Note that two different experimental dietary groups were then equally distributed (CT and AFB1).

Subsequently, two additional experimental conditions were established for each experimental diet (CT and AFB1 feeding)—(i) low stocking density (LSD, 4 g L^−1^), and (ii) high stocking density (HSD, 40 g L^−1^). Therefore, four duplicate treatments were tested—CT-fed group at LSD; CT-fed group at HSD; AFB1-fed group at LSD; and AFB1-fed group at HSD. All conducted treatments were carried out in duplicate for 10 days, keeping the above-described feeding regime. Note that in order to establish the HSD condition, fish were restrained in cages within the tank, maintaining in all replicates the same water volumes to ensure adequate and identical sanitary tank conditions. The experimental overcrowding conditions used here have previously been shown to activate the stress system in *S. aurata* [8,24,26,58,59,60].

All the experimental procedures complied with the guidelines of the University of Cádiz (Spain) and the European Union Council (2010/63/EU) for the use of animals in research, approved by the Ethics and Animal Welfare Committee from the Spanish Government (RD53/2013), and endorsed by the Regional Government (Junta de Andalucía reference number 28-04-15-241).

### 2.3. Sampling Protocol

In order to minimize tank effects, fish from experimental replicates were randomly sampled. Eight fish per treatment (from four different experimental treatments) were collected for sampling and subsequent analysis. Fish were netted and euthanized with a lethal dose of 2-phenoxyethanol (1 mL L^−1^ in seawater) and sampled. Blood was collected from the caudal peduncle into 1-mL syringes rinsed with a solution containing 25,000 U heparin ammonium salt (Sigma H6279) per 3 mL 0.9% NaCl. Plasma was obtained by centrifugation (4000× *g*, 3 min, 4 °C), frozen in liquid nitrogen, and stored at −80 °C until analysis. Fish were then beheaded and dissected. Liver and muscle biopsies were placed in empty Eppendorf tubes and snap-frozen in liquid nitrogen and stored at −80 °C until analysis. Moreover, biopsies of the head kidney, and whole pituitaries, were placed in individual Eppendorf tubes containing 500% (*v*/*w*) of RNAlater (Ambion^®^, Applied BioSystems, Austin, TX, USA). These samples were kept for 24 h at 4 °C and then stored at −20 °C until total RNA isolation was performed.

### 2.4. Plasma and Tissue Parameters

Cortisol levels were analyzed using the commercial kit Arbor Assays (Ann Arbor, MI, USA), “Enzyme Immunoassay kit” (Cortisol K003-H5). Osmolality was measured in 20 μL samples with a Fiske One-Ten vapor pressure osmometer (Fiske Associates, Advanced Instruments, Norwood, MA, USA). For assessing tissue metabolite levels, samples from liver and muscle were individually minced on an ice-cold Petri dish and subsequently homogenized by mechanical disruption (Ultra-Turrax^®^, T25basic with an S25N-8G dispersing tool, IKA^®^-Werke) with 7.5 vol. (*w*/*v*) of ice-cold 0.6 N perchloric acid and neutralized after adding the same volume of 1 M KHCO_3_. Subsequently, the homogenates were centrifuged (30 min, 3500× *g*, 4 °C), and the supernatants were recovered in different aliquots. The aliquots were then stored at −80 °C until used in metabolite assays. Metabolite concentrations in plasma (glucose, lactate, and triglycerides) and liver (glucose and triglycerides) were determined using commercial kits from Spinreact (Barcelona, Spain) (Glucose-HK Ref. 1001200; Lactate Ref. 1001330; Triglycerides ref. 1001311) with reactions adapted to 96-well microplates. Liver glycogen levels were assessed using the method from Keppler and Decker [61]. After subtracting free glucose levels, the glucose obtained after glycogen was determined using the commercial kit described above for glucose. Standards and samples were measured in duplicate. All the assays were run on an Automated Microplate Reader (PowerWave 340, BioTek Instrument Inc., Winooski, VT, USA) using KCjunior™ software.

### 2.5. RNA Isolation and Quantitative Real-Time PCR

The tissue biopsies were individually processed for total RNA extraction using NucleoSpin^®^ RNA kits (Macherey Nagel GmbH & Co. KG, Düren, Germany), except for pituitaries, for which NucleoSpin^®^ RNA XS kits were used (Macherey Nagel GmbH and Co. KG, Düren, Germany). An Ultra-Turrax^®^ T10 with a dispersing tool S10N-5G was used to homogenize both pituitaries and head kidneys. Genomic DNA (gDNA) was removed via on-column DNase digestion at 37 °C for 30 min using rDNase (RNase-free) included within the kits. The RNA concentration was measured with a Qubit 2.0 fluorimeter and Qubit RNA BR assay kit (Life Technologies, CA, USA), while RNA quality was tested in a Bioanalyzer 2100 with the RNA 6000 Nano kit (Agilent Technologies, Palo Alto, CA, USA). Only samples with an RNA integrity number greater than 8.0 were used for real-time quantitative PCR (qPCR). Total RNA (50 ng from the pituitary or 500 ng from the head kidney) from each sample was reverse-transcribed in a 20 μL reaction using the qScript™ cDNA synthesis kit (Quanta BioSciences, Gaithersburg, MD, USA) in a Mastercycler^®^ proS (Eppendorf, Hamburg, Germany), as previously described by Mata-Sotres et al., [62]. Reactions were diluted tenfold with 10 mM Tris-0.1 mM EDTA pH 8.0, to obtain final nominal concentrations of 2.5 ng μL^−1^ or 250 pg μL^−1^ for head kidneys or pituitaries, respectively. A pool of cDNAs from all the samples for each tissue was used for calibration plots, using six 1/10th serial dilutions from 10 ng to 100 fg (1 ng to 10 fg for pituitaries), to assess the linearity and efficiency of the different primer combinations, as well as for inter-assay calibration. Control reactions with RNase-free water (NTC) and RNA (NRT) were included in the analysis to ensure the absence of primer–dimers and genomic DNA contaminations. For all primer pairs, linearities (R^2^) and amplification efficiencies were in the ranges 0.997–1 and 0.900–1.052, respectively. Previously, primer pairs were tested for final working concentrations (optimum 200 nM) and temperature (60 °C). Reactions for qPCR were performed in triplicate with 10 or 1 ng of cDNA per well (estimated from the input of total RNA), forward and reverse primers (Table 1) for the named genes (200 nM each) and PerfeCTa™ SYBR^®^ Green FastMix™ (Quanta BioSciences Gaithersburg, MD, USA). Reactions were performed in a volume of 10 μL using Hard-Shell^®^, white well/blue shell, Low-Profile, Thin-Wall, 96-Well, Skirted, PCR plates (BioRad, Hercules, CA, USA) covered with Microseal^®^ B Adhesive Seals (BioRad). The thermocycling procedures were carried out with an initial denaturation and polymerase activation at 95 °C for 10 min, followed by 40 cycles of denaturation for 15 s at 95 °C, annealing and extension at 60 °C for 30 s, and finishing with a melting curve from 60 °C to 95 °C, increasing by 0.5 °C every five seconds. Melting curves were used to ensure that only a single PCR product was amplified and to verify the absence of primer–dimer artifacts. Relative gene expressions were quantified in a CFX Connect™ Real-Time PCR Detection System under the control and analysis of CFX Manager™ software (Bio-Rad) using the ΔΔCT method [63], corrected for efficiencies [64], and normalized by geometric averaging of two references genes [65], *eef1a* and *actb*. The genes were selected owing to their lower than 0.5 target stability M value and lower than 0.25 CVs (as indicated by BioRad CFX Manager Target Stability Value). A pool of all the cDNA samples was used as a calibrator on every qPCR plate to correct for inter-assay differences. Primers were used as previously described in Barany et al. [50].

### 2.6. Statistical Analysis

Statistical comparison for all given results was performed using a two-way ANOVA. All data are represented as the mean ± SEM (Standard Error of the Mean). Before these statistical analyses, both normality and equal variance were confirmed by Shapiro–Wilk, and D’Agostino–Pearson tests, respectively. In addition, for all data sets, outliers were identified by the ROUT method at Q = 1%. All two-way ANOVA analyses were followed by a Tukey’s post hoc test when significant differences were detected. All statistical analyses were performed with GraphPad Prism 6.0 (GraphPad Software, La Jolla, CA, USA), and significance for all tests was set at *p* < 0.05.

## 3. Results

### 3.1. Fish Mortality

No mortality was observed in any group throughout the duration of the experiment.

### 3.2. Plasma

Cortisol and glucose levels increased significantly in the CT-HSD group compared to the CT-LSD group, but not between the groups fed with AFB1 (AFB1-LSD vs. AFB1-HSD) (Figure 1 and Table 2, respectively). Triglycerides showed significantly lower levels in the AFB1 compared to the CT groups (-LSD and -HSD), but not between different stocking densities (-LSD vs. -HSD) within the same experimental diet (Table 2). No significant differences were detected in osmolality or lactate (Table 2). The significances of the two-way ANOVA for Figure 1 were—P_density_ = 0.62; P_diet_ = 0.04; P_interaction_ = 0.01.

### 3.3. Liver

Glucose was not significantly different within the CT or the AFB1 groups (-LSD vs. -HSD); however, significantly lower glucose levels were detected in the AFB1-HSD group than in the CT-HSD group (Figure 2A). Glycogen values decreased significantly (~half) in the AFB1 groups with respect to the CT groups at both stocking densities. In contrast, no significant differences were detected between different stocking densities (-LSD vs. -HSD) within the same experimental diet (CT or AFB1; Figure 2B). Triglycerides increased significantly more in the CT-HSD group than in the CT-LSD group, whereas they were unaffected within the AFB1 group (-LSD vs. -HSD; Figure 2C). The significances of the two-way ANOVA for Figure 2 were as follows—(A) P_density_ = 0.62; P_diet_ = 0.04; P_interaction_ = 0.01; (B) P_density_ < 0.01; P_diet_ < 0.0001; P_interaction_ = 0.67; (C) P_density_ = 0.20; P_diet_ = 0.46; P_interaction_ = 0.04.

### 3.4. Muscle

None of the analyzed parameters in this tissue showed significant differences between diets and/or stocking densities (Table 3).

### 3.5. Quantitative Real-Time PCR Expression

No significant differences were detected for hypophyseal *pomca1* or *pomca2* mRNA expressions. However, we observed a general increasing trend of *pomca1* expression analysis in both groups (-LSD and -HSD) fed with AFB1 compared to CT groups (Figure 3A). For *pomca2*, it tended to decrease only in the AFB1-HSD group compared to the AFB1-LSD, whereas the relationship was inverse between the CT groups (-LSD vs. -HSD).

Regarding *star* in head kidney, it significantly decreased in the AFB1-HSD compared to the AFB1-LSD group. Moreover, its mRNA expression’s starting point was significantly different between CT-LSD and AFB1-LSD (Figure 3C). The significances of the two-way ANOVA for Figure 3 were as follows—(A) P_density_ = 0.48; P_diet_ = 0.02; P_interaction_ = 0.95; (B) P_density_ = 0.65; P_diet_ = 0.41; P_interaction_ = 0.12; (C) P_density_ = 0.04; P_diet_ = 0.02; P_interaction_ = 0.07.

Expression analysis on *gh* mRNA enhanced significantly in the AFB1-HSD group compared to the AFB1-LSD group (Figure 4). The significances of the two-way ANOVA for Figure 4 were—P_density_ = 0.02; P_diet_ = 0.0005; P_interaction_ = 0.41.

## 4. Discussion

Our study revealed a significant increase in plasma glucose in CT-HSD, corresponding to the secondary stress response associated with an increase in plasma cortisol. Concomitantly, there was a decreasing hepatic glycogen trend, whereas free glucose increased. These changes suggest the stimulation of the glycogenolytic capacity due to cortisol action, as previously reported in *S. aurata* [66,67] under similar stress situations [8,25,58,59]. However, the group fed with AFB1 subjected to high stocking conditions (AFB1-HSD) did not increase plasma cortisol or plasma glucose. At the hepatic level, glycogen content significantly decreased in the AFB1 groups compared to the CT groups, whereas free glucose levels diminished instead of increasing. This change in hepatic carbohydrate reserve depletion was probably linked to prior aflatoxicosis background and then intensified by additional HSD exposure.

The differences observed in the CT and AFB1 groups subjected to LSD agree with Barany et al. [50], who demonstrated that chronic dietary AFB1 for the gilthead seabream affects the overall metabolic status, among other physiological effects. Remarkably, the triglyceride plasma levels were already lower in the AFB1–LSD than in the CT–LSD group due to the prior intake of AFB1 [50]. Moreover, the HSD condition did not induce further changes in plasma triglycerides in any of the groups. At the hepatic and muscular level, triglycerides showed an increasing trend in the CT-HSD group, whereas they decreased within the same tissues in the AFB1-HSD group. Consequently, this modification linked to previous AFB1 exposure could explain the later failure in cortisol enhancement (i.e., to mobilize lipids for steroidogenic synthesis) [68,69].

Furthermore, lactate, a pivotal element in the Cory cycle and anaerobic exercise, did not significantly differ among groups. Specifically, in the CT-HSD group, the lactate trend increased in both plasma and muscle compared to the CT-LSD group [70]. In contrast, in the AFB1-HSD compared to the AFB1-LSD group, lactate levels in plasma decreased. Although this observation was not statistically supported, the differences between dietary treatments might be related to a putative increase of the lactate dehydrogenase (LDH) activity, an enzyme involved in the interconversion of lactate in pyruvate and back, due to AFB1 effects [71]. In this sense, a higher LDH activity due to AFB1 would avoid plasma lactate level enhancement and a subsequent increase in reactive oxygen species production [45,71], which both might be in line with our general observations. Therefore, our findings show that the previous aflatoxicosis condition compromises metabolic orchestration to cope with the imposed HSD condition’s energetic demand, such as carbohydrate and lipid availability in plasma and liver. Consequently, reinforcing our hypothesis of a fundamental metabolic failure derived from a disrupted initial stress response in the AFB1-fed group [50], which is later intensified when exposed to additional overcrowding stress.

In this study, plasma cortisol levels significantly increased in the CT-HSD group compared to the CT-LSD group, as expected. This observation agrees with an activation of the stress system through hypercortisolemia (primary stress response) that induced the classical secondary and, subsequently, tertiary stress responses described previously in several fish species [5], including gilthead seabream [8,26,60]. In contrast, the AFB1-HSD group showed an altered response of the endocrine axis components. This modification in the physiological response included a decreasing trend for *pomca2*, while no plasma cortisol increase was detected. Remarkably, *star* enzyme expression was already higher in AFB1-LSD than in the CT-LSD group. The *star* enzyme is involved in cortisol synthesis and limits cortisol production by regulating cholesterol transfer within the mitochondria [72]. Interestingly, expression analysis on *star* mRNA in the head kidney in CT-HSD was not overexpressed, perhaps due to a negative feedback effect [73] of the high plasma cortisol levels observed in this group.

In contrast, and similarly to the inhibited *pomca2* trend after HSD exposure, *star* mRNA was already overexpressed in the AFB1-LSD group and significantly decreased at the HSD condition. These transcripts modifications might be similar to those caused by other mycotoxins, such as receptor binding competition [74] and/or inhibition of RNA/protein synthesis [75]. Further, as pointed out by Pottinger [76], and concerning our results, these dysregulations might also be due either to the direct effects of the toxicant (AFB1 or its subproducts) on endocrine glands and/or to a prolonged feedback suppression by corticosteroids.

The cortisol response to stressors could be altered in aquatic organisms previously exposed to pollutants. In this regard, fish such as the yellow perch (*Perca flavescens*) or Northern pike (*Esox Lucius*) caught in polluted areas produced 20–50% less plasma cortisol when subjected to subsequent overcrowding stress, showing their inability to increase serum cortisol in response to this specific stress [77,78]. An example of the impaired stress response in fish from polluted environments related to observable tissue-pathologies is the atrophy of pituitary corticotrope cells, which produce ACTH [79]. Unfortunately, the precise functional interference of cortisol by toxicants remains unknown, and it most likely depends on the nature of the toxicant and the magnitude of exposure of the organism to it [78]. Further studies assessing the pathway followed to corticosteroid synthesis under toxicant exposure would be required to answer this question.

In the gilthead seabream, the HSD condition has been shown to enhance *gh* expression [17]. Although we observed these increases, they were without significance between the CT groups. This discrepancy could be due to differences in the intensity and/or duration of the stress (70 g L^−1^ during 14 days in the previous work versus 40 g L^−1^ during 10 days in the present study). Moreover, dietary exposure to AFB1 under LSD conditions did not significantly increase *gh* mRNA expression, which has also been previously reported for this species [1]. However, *gh* expression was significantly up-regulated in the AFB1-HSD group, whereas it was not in the CT-HSD group. We suggest that this *gh* enhancement in the AFB1-HSD group could be related to the higher bioenergetic expenditure imposed by combining the two stressors—dietary AFB1 exposure plus crowding situation [2,19].

## 5. Conclusions

A dietary background of AFB1 alters the starting point of some parameters that subsequently seem to evoke failure in metabolic reorganization and stress response (at molecular and physiological levels). As a result, a previous aflatoxicosis might compromise an adequate and coordinated physiological response to cope with an additional stressor, such as confinement. In summary, this convergence of stressors does not affect survival, rendering the understanding of the threat problematic and ultimately potentially hinders a continuous deterioration of the reared fish farmed stocks subjected to standard aquaculture stressors.

## Figures and Tables

**Figure 1 animals-11-00753-f001:**
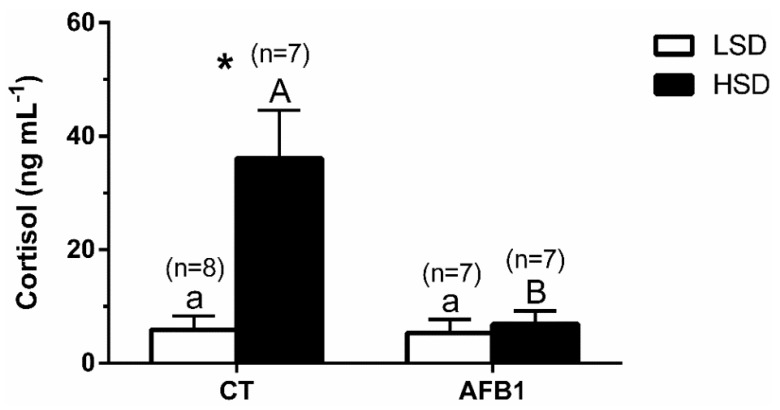
Changes in plasma cortisol levels in *S. aurata* individuals previously fed with different experimental diets (control, CT; with 2 mg aflatoxin B1(AFB1) kg^−1^ fish feed, AFB1) for 85 days and subsequently kept at different stocking densities (low stocking density (LSD), 4 g L^−1^; and high stocking density (HSD), 40 g L^−1^) for 10 days. Data are presented as mean ± SEM; *n* indicates the number of individuals. Different letters indicate significant differences (lowercase letters—within same LSD groups fed with different experimental diets; capital letters—within same HSD groups fed with different experimental diets). Asterisks (*) indicate significant differences within an identical experimental diet, between different stocking density groups (*p* < 0.05, two-way ANOVA followed by Tukey’s post hoc test).

**Figure 2 animals-11-00753-f002:**
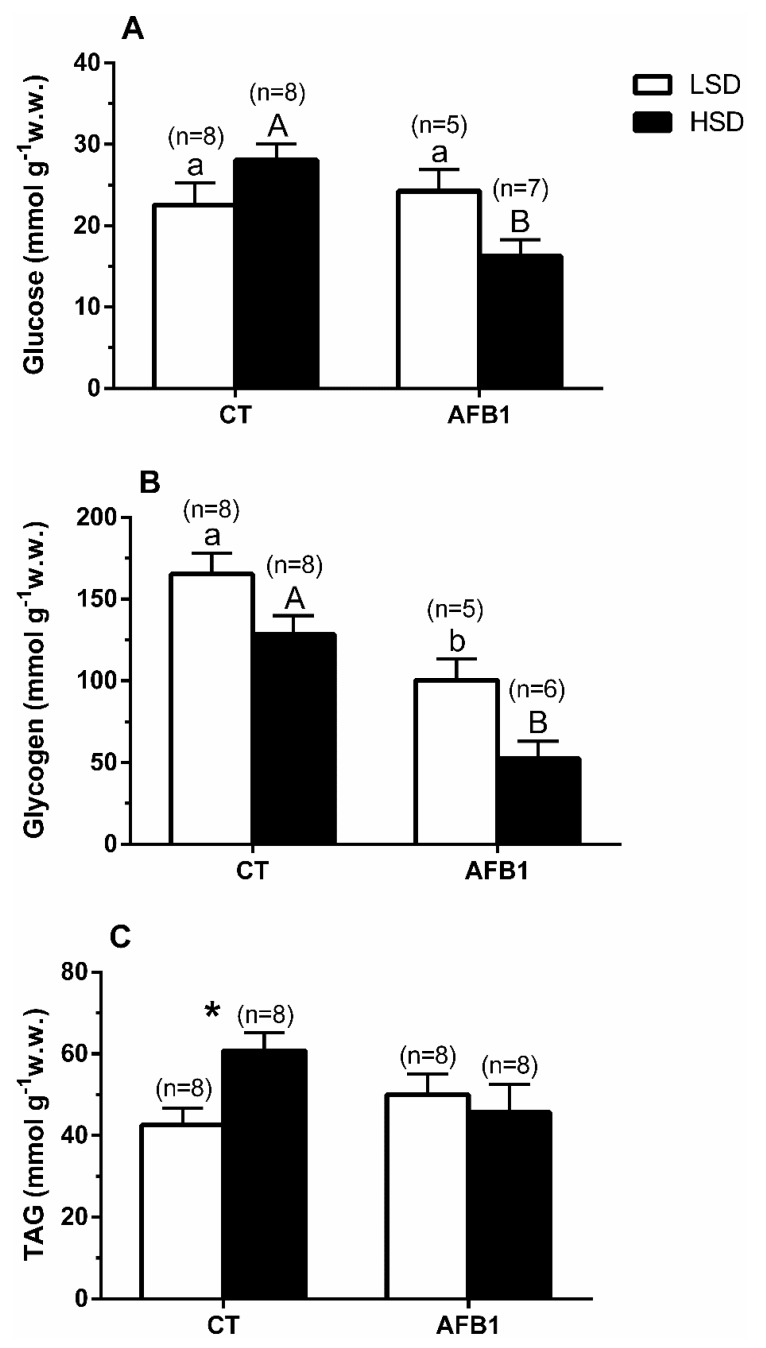
(**A**) Changes in liver glucose, (**B**) liver glycogen, and (**C**) liver triglyceride levels in *S. aurata* individuals previously fed with different experimental diets and subsequently kept at different stocking densities. Data are presented as mean ± SEM. Values are expressed as grams of the wet weight of tissue (g^−1^ w.w.). Further details in legend Figure 1.

**Figure 3 animals-11-00753-f003:**
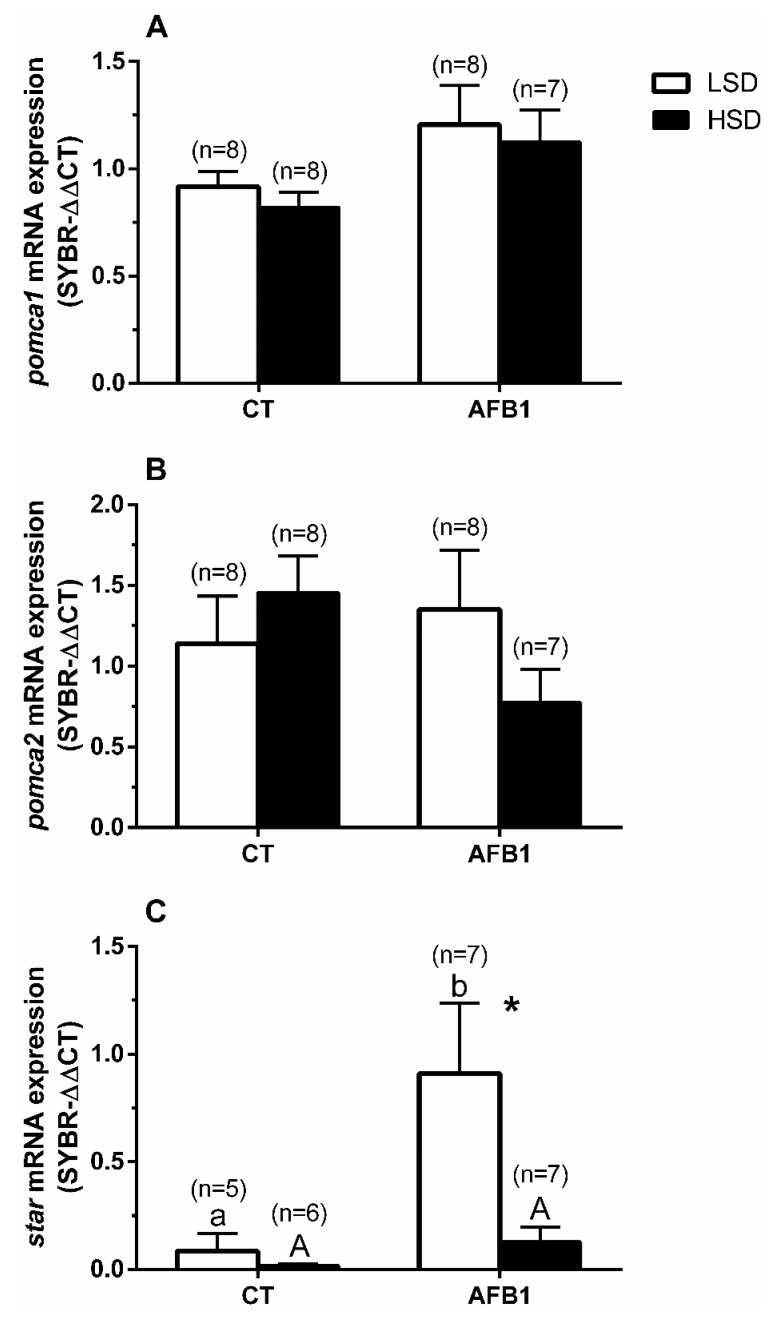
Changes in mRNA expression levels (relative to *eef1a* and *actb*) of (**A**) hypophyseal *pomca1*, (**B**) *pomca2*, and (**C**) *star* in the head kidney in *S. aurata* individuals previously fed with different experimental diets and subsequently kept at different stocking densities. Data are presented as mean ± SEM. Further details as in legend from Figure 1.

**Figure 4 animals-11-00753-f004:**
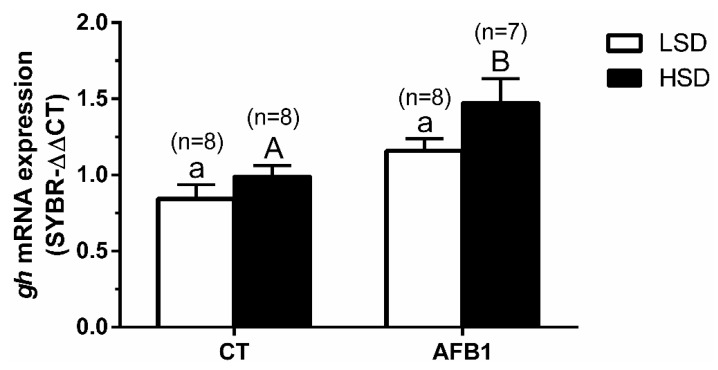
Changes in mRNA expression levels (relative to *eef1a* and *actb*) of hypophyseal *gh* in *S. aurata* individuals previously fed with different experimental diets and subsequently kept at different stocking densities. Data are presented as mean ± SEM. Further details as in legend from Figure 1.

**Table 1 animals-11-00753-t001:** Specific primers used for real-time qPCR expression analysis and sizes of the amplified products.

Primers	Nucleotide Sequence (5′→3′)	GenBank acc. no.	Amplicon Size (bp)
*pomca1* Fw	AGCCAGAAGAGAGAGCAGTGAT	HM584909	122
*pomca1* Rv	ATCGGGTCAGAAAACACTCA		
*pomca2* Fw	AGCTCGCCAGTGAGCTGT	HM584910	81
*pomca2* Rv	CCTCCTGCATCACTTCCTG		
*gh* Fw	CGTCTCTTCTCAGCCGAT	U01301	131
*gh* Rv	GCTGGTCCTCCGTCTGC		
*star* Fw	GAGCACAGATGTTGGCTTCA	EF640987	175
*star* Rv	GAAACAATCGAGGCACACAA		
*eef1a* Fw	AGAGGCTGTCCCTGGTGA	AF184170	137
*eef1a* Rv	TGATGACCTGAGCGTTGAAG		
*actb* Fw	TCTTCCAGCCATCCTTCCTCG	X89920	108
*actb* Rv	TGTTGGCATACAGGTCCTTACGG		

Gene abbreviations: *pomca1*: proopiomelanocortin a1; *pomca2*: proopiomelanocortin a2; *gh*: growth hormone; *star*: steroidogenic acute regulatory protein; *eef1a*: elongation factor 1-alpha; *actb*: beta (β)-actin.

**Table 2 animals-11-00753-t002:** Plasma parameters in *S. aurata* individuals previously fed with different experimental diets and subsequently kept at different stocking densities. Data are presented as mean ± SEM; *n* indicates the number of individuals. Further details in legend Figure 1.

Parameters	CT-LSD	CT-HSD	AFB1-LSD	AFB1-HSD
Osmolality (mOsm kg^−1^)	363 ± 4 (*n* = 8)	345 ± 7 (*n* = 7)	361 ± 8 (*n* = 7)	362 ± 22 (*n* = 7)
Glucose (mM)	3.64 ± 0.19 ^a^ (*n* = 8)	4.42 ± 0.24 ^A^* (*n* = 8)	3.64 ± 0.15 ^a^ (*n* = 8)	3.68 ± 0.28 ^A^ (*n* = 8)
Lactate (mM)	1.28 ± 0.12 (*n* = 8)	1.66 ± 0.13 (*n* = 8)	1.92 ± 0.27 (*n* = 8)	1.80 ± 0.39 (*n* = 8)
Triglycerides (mM)	1.02 ± 0.07 ^a^ (*n* = 8)	1.01 ± 0.06 ^A^ (*n* = 8)	0.46 ± 0.07 ^b^ (*n* = 8)	0.42 ± 0.09 ^B^ (*n* = 8)

The significances of the two-way ANOVA for Table 1 were as follows—(Osmolality) P_density_ = 0.46; P_diet_ = 0.53; P_interaction_ = 0.42; (Glucose) P_density_ = 0.07; P_diet_ = 0.10; P_interaction_ = 0.10; (Lactate) P_density_ = 0.63; P_diet_ = 0.14; P_interaction_ = 0.31; (Triglycerides) P_density_ = 0.73; P_diet_ < 0.0001; P_interaction_ = 0.42.

**Table 3 animals-11-00753-t003:** Muscle parameters in *S. aurata* individuals previously fed with different experimental diets for 85 days and subsequently kept at different stocking densities. Data are presented as mean ± SEM. Values are expressed as grams of the wet weight of tissue (g^−1^ w.w.). Further details in legend Figure 1.

Parameters	CT-LSD	CT-HSD	AFB1-LSD	AFB1-HSD
Glucose (mmol g^−1^ w.w.)	13.75 ± 1.67 (*n* = 8)	13.02 ± 2.86 (*n* = 8)	14.83 ± 3.36 (*n* = 8)	11.68 ± 2.24 (*n* = 8)
Glycogen (mmol g^−1^ w.w.)	1.22 ± 0.39 (*n* = 5)	1.49 ± 0.62 (*n* = 5)	1.06 ± 0.17 (*n* = 8)	1.93 ± 0.57 (*n* = 8)
Lactate (mmol g^−1^ w.w.)	38.70 ± 2.30 (*n* = 8)	44.30 ± 2.13 (*n* = 8)	32.61 ± 2.30 (*n* = 8)	33.85 ± 4.37 (*n* = 8)
Triglycerides (mmol g^−1^ w.w.)	16.91 ± 2.13 (*n* = 8)	20.83 ± 3.11 (*n* = 8)	19.02 ± 1.84 (*n* = 8)	16.16 ± 3.24 (*n* = 8)

The significances of the two-way ANOVA for Table 2 were as follows—(Glucose) P_density_ = 0.70; P_diet_ = 0.43; P_interaction_ = 0.59; (Glycogen) P_density_ = 0.28; P_diet_ = 0.79; P_interaction_ = 0.57; (Lactate) P_density_ = 0.25; P_diet_ < 0.01; P_interaction_ = 0.46; (Triglycerides) P_density_ = 0.84; P_diet_ = 0.63; P_interaction_ = 0.21.

## Data Availability

Data generated in this study are available on request from the corresponding author (Andre Barany; email: andre.barany@uca.es).

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
