# Peer review of "Aflatoxicosis Dysregulates the Physiological Responses to Crowding Densities in the Marine Teleost Gilthead Seabream (Sparus aurata)"

_animals, 2021, doi:10.3390/ani11030753_

Round 1

Reviewer 1 Report

Letter to Authors
animals-1094018
Aflatoxicosis dysregulates the physiological responses to crowding densities in the marine teleost gilthead seabream (Sparus aurata)
Andre Barany, Juan Fuentes, Gonzalo Martinez-Rodriguez, Juan Miguel Mancera

210119

Dear authors,
Your MS is potentially interesting posting a novel insights on detrimental consequences of stressful environments in aquaculture practices. It is thus worth publishing in an international journal like Animals. This MS is, however, not well-written. Introduction and discussion sections are complicated disturbing readers' understanding. Though I clicked "minor" this time, revision will be substantial. I will read a prospecting revised version.
See below for detail.

L13 simple summary
This section should be simpler.

L13-16
In a previous study, .. remained unaffected. -> A fungal toxin, aflatoxin B1 (AFB1) undermines growth and stress axes of gilthead seabream (Sparus aurata) with depletion of [somatic] carbohydrate and lipid reservoirs.

L18
during 10 days -> delete

L19
for 85 days -> delete

L19-22
Hence, we found .. culture practice. -> AFB1 caused physiological and molecular dysfunctions in response to crowding densities.

L22
These -> Our

L34
star mRNA ?

L39 keywords
aflatoxicosis, crowding -> replace
Do not list words which appear also in the title. Duplicate hits upon computer search do not make sense. Give words that do not appear in the title to draw attention from wider readership. Posting words that neither appear in the abstract is better, because even in full-text search/indexing robots may not weigh much on words deeper (posterior) in the text. Hint: aquaculture, Sparidae, Perciformes, carbohydrate, lipid, endocrine related genes, etc.

L43
[2]
References should be in numbers of appearance in the text. Ref item #1 appears in L91 just before #30, and then renumbering is necessary rotating (circular shift) #1 through #29 both in the main text and the list.

L44
is a need to improve (wordy) -> improves
and -> delete

L49
and -> delete

L51
dead -> {death, mortality}

L54
when -> which

L63
The fact that .. worldwide (wordy) -> The worldwide {decline, decrease} of wild fishery-stocks for fishmeal production
Use noun phrases for conciseness.

L72
In this regard -> In this regard, (insert a comma)
a fact ? -> a serious problem ?
The citing literature is not regarding fish feeds, and this not a fact.

L78
spp. (English) -> in Roman

L84
, ROS -> (ROS)
Use consistent style presenting abbreviations.

L85
deadly -> fatal ?

L95
For human consumption, .. can be deadly [34]. (jumping topics) -> delete

L96
aflatoxin levels -> aflatoxin levels for human and livestock ?

L105
Add statements regarding choice of metabolites and genes for examination.

L111
(CT) -> (control, CT)?

L119
furcal -> fork ?

L127
previously described feeding regime -> feeding regime as above ?

L144
representative ?
It is unclear how many individual tissue samples were taken. In the previous paragraph, readers will understand that all fish under CT/AFB1 and LSD/HSD conditions were scarified and biopsied. Then readers encounter this dubious word. You should mention number of fish chosen (randomly?) and examined at the top of this subsection 2.2.

L148
8 -> eight

L155
samples from liver and muscle
Pooled by each experimental condition?

L213
Add a footnote on spelled out names of the genes.

L216,237,259,266
SEM -> SE ?

L237,259,282,285
n=6-8,5-8,7-8
Quite unclear! You should put number of individuals tested on or under each bar of the figure picture.

L244,258,265
for 85 days and subsequently confined (redundant) -> delete

L249,251
group -> groups

L251
no significant differences were detected ?
I can see differences in height of bars both on control and AFB1 (a vs A and b vs B). You may present P values instead just like L273.

L253,277,334
Lastly, (verbose) -> delete

L255
unaffected within the AFB1
You may present the P value.

L257
TAG -> triglyceride

L268
expression -> {analysis, expression analysis}

L288-309
The gilthead seabream is .. growth axis was altered [1]. -> move to introduction section around L88-105. Some revision may be necessary to fit in. See also a comment on L105.

L298
response -> responses
this species' physiological responses -> physiological responses of this species

L311
reveals -> revealed

L318-320
decreased about 2-fold .. 47.64 % and 22.29 %, respectively
Results appeared here at the first place! Move to the result section. Make concise here using noun phrases like "significant decrease of glycogen content ..".

L320,330,387
Thus, suggesting ..
Not complete sentences. -> Noun phrases (see above) suggest ..

L352-360
Cortisol is .. in muscles [59]. -> move to introduction section around L88-105. Some revision may be necessary to fit in. See also a comment on L105.

L374-383
The pituitary gland acts as .. stress between diets. -> move to introduction section around L88-105. Some revision may be necessary to fit in. See also a comment on L105.

L394
carp -> perch

L401
And as pointed out by -> {Further as pointed out by, As pointed out further by}
Do not begin a sentence with "and".

L406
degree -> {extent, strength} ?

L406,414,419
exposition -> exposure ?

L409-410
GH is .. including gilthead seabream [73]. -> move to introduction section around L88-105. Some revision may be necessary to fit in. See also a comment on L105.

L424
a not AFB1 fed CT group -> a CT group without AFB1 intake

L426
In summary, (verbose) -> delete
Why did you like to insert these words here? Did you feel discomfort about your story flow? If so, you should also revise over here for better story flow. The best story flow does not need connecting words.

L448 references
Check the reference list carefully again from the beginning. Reference lists are frequently hotbeds of errors. You might add, omit or swap citation in the main text on the way internal revision. You will rotate several items (see L43). Numbering of the references might then shift. If so, readers think you are making irrelevant citation. It is the authors' responsibility that all references are properly cited.

Reviewer 2 Report

The study assessed the effects of aflatoxin B1 (AFB1) on the physiological consequences of high stocking density and low stocking density during 10 days in seabream juveniles. It sounds interesting. However, the writing needs to be improved. The authors need to find a native English Speaker to help revise the paper. The detailed comments are below.

Major comments:

  1. Writing is hard to understand and need to be revised by a native English Speaker.
  2. This is a 2 by 2 factory experimental design, all the statistical analyzed results need to contain the main effects of AFB1, density and their interaction. The way of express the results in all the tables and figures looks not right.

Minor comments:

  1. Abstract, the experimental design did not described clearly here. How many replicates have been used in each treatment, etc.?
  2. Line 31, please remove the [1] from the abstract.
  3. Lines 72-85, please add the latest references about the contamination of AFB1 and metabolism. Such as: 1) Individual and Combined Occurrence of Mycotoxins in Feed Ingredients and Complete Feeds in China. Toxins. 2018; 2) Aflatoxin B1 metabolism: Regulation by phase I and II metabolizing enzymes and chemoprotective agents. Mutation Research-Reviews in Mutation Research. 2018.
  4. Line 110, Use “MJ kg-1 ” instead of “mJ kg-1”.
  5. Lines 118-124, the writing of experimental design was very hard to understand. This is a 2 by 2 factory experimental design. The authors need to reference some publications and learn how to write it.
  6. Line 117-135, Does the Animals and experimental designs consider the influence of water temperature? And What is the water temperature in the test?
  7. Line 212, please add the footnote for the table 1. Such as the full name for the abbreviations. Please check all the rest tables.
  8. Line 224 and 226, “3.1. Fish mortality” and “3.1. Plasma” should be “3.1. Fish mortality” and “3.2. Plasma”. Please also check these in the main text.
  9. Table 1. Three-line tables should be use. Please check second line.
  10. Line 234, the way to displayed the results looks not right. Using both upper and lower case of letter to express the different for a 2 by 2 factor experiment design is not right. Please check all the data.
  11. Figure 2. Use “CT” instead of “control”.

Round 2

Reviewer 1 Report

Letter to Authors
animals-1094018-v2
Aflatoxicosis dysregulates the physiological responses to crowding densities in the marine teleost gilthead seabream (Sparus aurata)
Andre Barany, Juan Fuentes, Gonzalo Martinez-Rodriguez, Juan Miguel Mancera

210209

Dear authors,
You might pay efforts improving your MS, but it is not enough. Your revision introduced several new issues. Extensive English proofreading is absolutely necessary. The very 1st sentences of the simple summary and introduction sections disheartened me. See below for detail.

Do not use hyphenation. It will cause typesetting errors. Check your usage of tense. Make consistent in using abbreviations. Reference list is not corrected. You seem dishonest.

L15
[somatic]
Bracketing words commonly means options for omission/inclusion.

L19
Subsequently, rebounding on .. industry (not a complete sentence) -> That risk subsequently rebounds on .. industry

L24
supplementation with -> administration of

L27
indicate -> indicated
Results should be in past tense throughout.

L30
analyzed expression on -> expression of

L33
confirm -> confirmed

L40
The most common aquaculture practice stressors (does not make sense) -> The most common stressors in aquaculture [practices]

L42
Especially in aquaculture activity (redundant) -> delete[

L43
profitability -> {economy, economic profitability}

L45,391
stress system -> stress-{control, managing} system

L46
releases (uncountable) -> release
"A release" is OK.

L60
proopiomelanocortin (POMC) -> POMCs
See L57

L62
MSH -> melanocyte-stimulating hormone (MSH)

L76
in -> of

L82
the hypothalamic-pituitary-interrenal (HPI) -> HPI

L87
aquaculture production's main profit -> the major profit of aquaculture [production]

L96
arise -> have arisen ?

L110
(ROS) (appears one more time only) -> delete

L122
strictest -> most strict

L130
aflatoxin B1 (AFB1) -> AFB1
See L105.

L136
axis -> axes

L138
a well-known farmed biological model [56] (redundant) -> delete

L156
Juvenile individuals (wordy) -> Juveniles

L277,292,303,320,331,335
results -> {significance, probabilities}

L279,291,306,319,337,342
Add "n=" to every numbers in parentheses for better understanding. Instead of this addition, omit over-explanatory sentences at L283,290,309,317,340,344.

L359
Indicating that (not a complete sentence) -> This {change, modification} indicates

L365
previous diet with -> {preceding, prior} intake of

L372
a classic indicator .. demanding situations [70] (wordy) -> a pivotal element in the Cory cycle and anaerobic exercise [70]

L376
[70] (irrelevant citation) -> delete

L381
ROS -> reactive oxygen species

L394
AFB1 fed group exposed to HSD -> AFB1-HSD

L417
will depend -> depends

L458 references
Check the list carefully again from the beginning. Find out and correct many errors. You seem not honest.

L459,468,480,etc (many)
Book chapter titles should be in lower case just like paper titles.

L470,495
Oncorhynchus mykiss -> in Italics
Journal?

L475,478,499,etc (many)
Sparus auratus -> in Italics

L483
Oreochromis mossambicus -> in Italics

L486
Verasper moseri -> in Italics

L501,636
Paper titles should be in lower case.

L504,516,etc
Sparus aurata -> in Italics
Journal?

L521
Incomplete paper title.

L527
Dicentrarchus labrax -> in Italics

and more ..

Reviewer 2 Report

No further comments.

Author Response

Dear Reviewer, thanks for your revision.

Round 3

Reviewer 1 Report

Letter to Authors
animals-1094018-v4
Aflatoxicosis dysregulates the physiological responses to crowding densities in the marine teleost gilthead seabream (Sparus aurata)
Andre Barany, Juan Fuentes, Gonzalo Martinez-Rodriguez, Juan Miguel Mancera

210224

Dear authors,
You still need to revise your MS CAREFULLY. I am upset to see this CARELESSLY revised MS. Use of abbreviations is complicating and inconsistent. See below for detail.

L41
by inferring (does not make sense) -> [by] {impairing, damaging, disturbing}

L42
in this activity -> {among those {practices, activities}, concerning those {practices, activities}, regarding those {practices, activities}}
What is "this"?

L56
adrenocorticotropic hormone release (ACTH) -> adrenocorticotropic hormone (ACTH) release

L83
Which ultimately .. (not a complete sentence) -> continue to the previous sentence

L85
the organism's death (wordy) -> mortality

L138
an 138 important farmed species [56] (redundant) -> delete
See L129.

L139
aflatoxin B1 -> AFB1

L163
What is D2? Reference(s) may be needed when it is a specific mode of dose.

L167
AFB1 -> D2-AFB1 (throughout)
You have introduced a new term in L163. Then you should use it throughout consistently.

L169
high stocking density -> HSD

L283
low stocking density, -> delete
high stocking density, -> delete

L284
This self-explanatory matter is unnecessary.

L286
low stocking density -> LSD

L287
high stocking density -> HSD

L347
the control group under high stocking density (CT-HSD) -> CT-HSD

L352
a dietary dose ? -> D2 ?
then ..
the group .. conditions (AFB1-HSD) -> D2-AFB1-HSD group

L359
low stocking density (-LSD) -> LSD

L369
exercise -> exercise, (insert a comma)

L374
lactate dehydrogenase activity (LDH) -> lactate dehydrogenase (LDH) activity
